# Child undernutrition is associated with maternal mental health and other sociodemographic factors in low-income settings in Dhaka, Bangladesh

Kazi Muhammad Rezaul Karim[1]*, Md Hafizul Islam[1], Tasmia Tasnim[2], Sumaiya Akter[2]

**1** Institute of Nutrition and Food Science, University of Dhaka, Dhaka, Bangladesh, **2** Department of Nutrition and Food Engineering, Daffodil International University, Daffodil Smart City, Birulia, Savar, Dhaka, Bangladesh

* rkarim98@gmail.com, rezaul.infs@du.ac.bd

## Abstract

### Background

Maternal mental health and other underlying factors might affect a child's nutritional status. This study assesses child undernutrition and its associated characteristics, including maternal mental health, in low-income settings in Dhaka, Bangladesh.

### Methods

A community-based cross-sectional study was conducted among 397 lactating mothers with children aged 6–23 months from low-income settings in Dhaka. Anthropometric measurements were taken following standard protocols, and Z-scores for weight-for-age, height-for-age, and BMI-for-age were calculated. Maternal depression and anxiety were assessed using the Patient Health Questionnaire-9 and the Generalized Anxiety Disorder 7-Item Scale, respectively. The child feeding index was developed based on breastfeeding, dietary diversity, and meal frequency. Multivariate logistic regression models explored the relationship between child undernutrition and maternal mental health and other risk factors.

### Results

In low-income regions of Dhaka, the prevalence was 31.9% for stunting, 14.0% for wasting, and 24.1% for underweight children. Approximately half of the mothers experienced depression (55%) and anxiety (50%). High maternal depression levels were associated with increased odds of stunted (AOR = 1.80, 95% CI = 1.10–2.94, p < 0.05) and wasted (AOR = 2.70, 95% CI = 1.38–5.28, p < 0.05) children. Similarly, anxiety was linked to a higher risk of underweight children (AOR = 1.77, 95% CI = 1.04–3.11, p < 0.05). Female children had approximately twice the risk of stunting than boys

**Data availability statement:** All relevant data are within the manuscript and its Supporting Information files.

**Funding:** The project was completed under the research grant scheme for public university teacher (2022-23) that is financially supported by the University Grant Commission (UGC), Bangladesh. The funders had no role in study design, data collection and analysis, decision to publish, or preparation of the manuscript. There was no additional external funding received for this study

**Competing interests:** The authors have declared that no competing interests exist.

(AOR = 2.13, 95% CI = 1.32–3.44, p < 0.01). Younger maternal age also doubled the risk of stunting (AOR = 1.97, 95% CI = 1.20–3.22, p < 0.01). Low adherence to a feeding index increased the odds of stunting (AOR = 3.21, 95% CI = 1.99–5.16, p < 0.001) and underweight (AOR = 4.20, 95% CI = 2.50–7.07, p < 0.01). Children born to underweight mothers were almost twice as likely to become underweight (AOR = 2.01, 95% CI = 1.01–4.03, p < 0.05) compared to those born to normal/overweight mothers.

## Conclusion

Maternal depression and anxiety adversely affect the nutrition of their children. Sociodemographic factors such as the child's sex, maternal age, maternal health, and child feeding practices significantly contribute to child undernutrition. Policy initiatives should prioritize maternal mental health and address child undernutrition in these settings.

## Introduction

Child undernutrition is a global concern, particularly in low- and middle-income countries (LMICs), including Bangladesh. Globally, among children under five years, 149 million are stunted (low height-for-age), 45 million are wasted (low weight-for-height), and 37 million are underweight (low weight-for-age) [1]. These dimensions of undernutrition along with low birth weight are leading causes of child mortality, morbidity, cognitive impairment, and growth retardation worldwide [2]. Like many Asian countries and other LMICs, the prevalence of child undernutrition is still a serious issue in Bangladesh albeit the prevalence reduced over the years here. According to the latest Bangladesh Demographic and Health Survey (BDHS), 2022, the prevalence of stunting, wasting, and underweight children under five years of age was 24%, 11%, and 22%, respectively and the prevalence was higher in the rural and low-income households [3]. Similarly, in low-income regions and cities, the prevalence of child undernutrition is higher than in non-slum urban areas [4]. Different factors contribute to child undernutrition in Bangladesh, including food insecurity, poverty, feeding practices, care, maternal education, empowerment, nutritional knowledge, physical health, and mental health [5–9].

According to UNICEF's conceptual framework of child malnutrition, maternal care, and feeding practices for their children are two crucial underlying and immediate factors of child health [10]. These factors are substantially impacted by the mental health of the mothers. Research has consistently shown that maternal mental health plays a crucial role in shaping various aspects of child development [11–15]. These shreds of evidence suggest that poor child growth was found when maternal depression and anxiety existed. When a mother experiences inadequate mental well-being, it can lead to suboptimal breastfeeding and complementary feeding practices, inadequate childcare, fewer hygiene practices for the child, reduced mother-child interactions, and an increased likelihood of emotional and medical issues in children [11–13,15]. Depressed mothers may engage less in stimulating interactions and

struggle with maintaining hygiene practices, increasing risks of diarrheal diseases and infections, leading to the children being undernourished. Their depression and anxiety can reduce motivation and confidence in seeking timely healthcare, leading to missed vaccinations, delayed illness treatment, and inadequate growth monitoring. Moreover, they may struggle with breastfeeding initiation, duration, and exclusivity, leading to early cessation and inadequate nutrition. Anxiety and mental distress may result in poor dietary diversity and inconsistent feeding patterns. Similarly, the feeding practices of the children are compromised due to food insecurity and poverty in Bangladesh, leading to the prevailing child undernutrition [16]. These pathways highlight the importance of integrating maternal mental health support into child nutrition programs, particularly in low-income settings.

Several studies identified the socio-economic determinants of child undernutrition in Bangladesh at national and subnational levels [5–9,16]. Urban areas in Bangladesh encompass a diverse population in terms of distinct economic disparity and lifestyle which might affect the nutrition of children [17]. A few studies were also conducted in urban slum areas to explore child undernutrition and its contributing factors [9,18,19]. However, the non-slum low-income regions in cities like Dhaka remain unexplored in terms of child undernutrition and their determinants. Unlike slum dwellers, who often rely on government or NGO-led interventions, non-slum low-income families may have limited access to structured support systems, despite experiencing similar economic hardships. Their housing conditions, employment stability, access to healthcare, and food security differ from both slum populations and middle-income urban residents, making them a unique yet overlooked group in child nutrition research. Understanding the specific challenges faced by these communities is crucial for designing targeted interventions that address their nutritional vulnerabilities. Moreover, while studies linking maternal mental health and child undernutrition are scarce, no previous research has specifically examined this relationship in non-slum low-income urban settings.

In the present study, we examined child undernutrition and maternal mental health in low-income settings in Dhaka, Bangladesh. We also assessed the associations of maternal mental health and other background characteristics with child undernutrition. This finding would provide an in-depth understanding of the factors of child undernutrition for policymakers, guiding decisions to prevent child undernutrition.

## Materials and methods

### Study design and setting

A community-based cross-sectional study was conducted among lactating mothers having children aged 6–23 months to explore nutritional status, child feeding practices, and their sociodemographic determinants. The children were from low-income settings in Hazaribagh and Mohammadpur thanas (police stations) (except Geneva camp) of Dhaka south and north city corporations, Bangladesh. This low-income group maintains livelihoods as day laborers, rickshaw pullers, owners of small-size self-businesses, or other low-salaried workers (e.g., garment employees). The study was conducted from October 2023 to January 2024.

### Sample size and sampling technique

The study population consisted of mothers and their children aged 6–23 months from low-income settings in Dhaka city. We conveniently surveyed 423 children and their mothers for this study. A total of 397 mother-child pairs were interviewed in the study and 26 mothers refused to be a part of the study. The sample size was calculated by the formula $n = z^2 pq/d^2$. Where n is the intended sample size, z is the level of significance at the 95% CI (=1.96), p is the predicted proportion of an attribute in the population, q is 1–p, and d is the desired degree of precision, typically set at 0.05. A prior study conducted in an urban slum region in Bangladesh [9], where the prevalence of common mental disorders was 46.2% and stunting was 44.3%, we assumed a sample power of 50%. With a precision of 5% and a confidence interval of 95%, the minimum sample size required was 384.

 

**Inclusion and exclusion criteria.** Participants were eligible for inclusion in the study if they met the following criteria:

- Lactating mothers with children aged 6–23 months.

- Permanent residents of the selected low-income areas in Hazaribagh and Mohammadpur thanas, Dhaka.

- Primary caregivers responsible for the child's feeding and care.

- Provided informed consent to participate in the study.

Participants were excluded from the study if they met any of the following criteria:

- Mothers who were not the primary caregivers of the child.

- Mothers with severe illnesses or disabilities that could interfere with their ability to provide information.

- Children with congenital anomalies or severe chronic illnesses that could influence their growth and nutritional status.

## Data collection

Face-to-face interviews were conducted with 397 mothers having children aged below 2 years in low-income settings in selected areas. A structured questionnaire comprising different sections (e.g., socio-demographic and socio-economic information, maternal background and health services-related information, and infant and young child feeding (IYCF) related information) was prepared. The enumerators were exhaustively trained in data collection.

## Outcome variables

**Anthropometric measurements of children and mother.** Anthropometric measurements were conducted by trained professionals following standard WHO procedures to ensure accuracy and reliability. Children's weight was measured using a SECA scale with an accuracy of 0.1 kg. Height or length was measured using a wooden height-length board, which was calibrated against a standard anthropometric scale, with an accuracy of 0.1 cm. For children aged 6 months to 2 years, recumbent length was measured, ensuring that the child was lying straight. For Mothers height was measured using a wooden height board, following standard anthropometric procedures. The WHO Anthro Software (version 1.0.4, 2009) has been used to compute nutrition indices such as the Z-scores for weight-for-age (WAZ), height-for-age (HAZ), and BMI for-age-Z score (WHO, 2009). Stunting, underweight, and wasted among children were defined as HAZ, WAZ, and BMI-for-age -Z score, less than -2 SD below the median of the WHO reference population, respectively. Maternal weight and height were used to calculate BMI ($kg/m^2$). Maternal undernutrition was defined as BMI < 18.5 $kg/m^2$.

**Assessment of maternal mental health.** Nine questions on the Patient Health Questionnaire-9 (PHQ-9) gauged respondents' levels of depression. Each question had a score ranging from 0 to 3 based on the response: 0 for "not at all," 1 for "several days," 2 for "half of the days," and 3 for "nearly every day." The cumulative score ranged from 0 to 27 points. Cumulative scores <5, 5–9, 10–14, and 15–27 for no depression, mild depression, moderate depression, and severe depression, respectively, were used to categorize the severity of depression [20]. In this study, the reliability of the PHQ-9 scale was evaluated using Cronbach's alpha. We obtained a value of 0.807 which indicated good internal consistency.

Seven fundamental questions comprised the Generalized Anxiety Disorder 7-Item Scale (GAD-7), which assesses respondents' anxiety disorders. The overall score varied between 0 and 21. Depending on the response, each question had one of four possible scores: 0 (not at all), 1 (several days), 2 (more than half the days), or 3 (nearly every day). With four distinct segments, a greater score denoted a higher level of anxiety; <5, 5–9, 10–14, and 15–21, respectively, indicated no, mild, moderate, and severe anxiety [21]. The reliability of the GAD-7 scale was evaluated using Cronbach's alpha (0.854) which indicated good internal consistency.

## Independent variables

**Child feeding index.** Following a previous study, the child feeding index was created [22]. Meal frequency, dietary diversity, breastfeeding, and the use of baby feeding during the previous 24 hours were the variables included in the index's composition. For this study, an IYCF index was developed using age-specific feeding guidelines (6–8 months, 9–23 months). Children in all age groups who were breastfed received two points, while those who were not received zero. One point was awarded for not using a bottle, and zero for using one. Using a list of seven food groups, dietary diversity was assessed by counting the number of food groups a child consumed in the previous 24 hours: eggs; pro-vitamin A-rich foods (yellow and orange-fleshed roots and tubers, orange-fleshed fruits, and dark green leafy vegetables); legumes; nuts and seeds; dairy products; flesh foods (meat, fish, and poultry); and other fruits and vegetables [23].

A feeding index was created using a modified version of a published diet diversity scoring system. For all age groups, eating from four or more food groups obtained two points, eating from one to three food groups obtained one point, and children who did not eat from any food group during the previous 24 hours received zero points. Children aged 6–8 months and 9–23 months who were fed at least twice and three times, respectively, received two points based on the required meal frequency. Fewer meals were worth one point, and none at all. A final IYCF index was created by adding the scores for each of the variables mentioned, taking insight from a previous study [24]. Scoring technique for the child feeding index is given in S1 Table. The 50th percentile was used to figure out the feeding scores and classify them as "Low" or "High".

**Sociodemographic variables.** Child-, maternal-, and household-level variables were selected based on previously published studies as the independent variables [5,6,25–30]. Child-related variables included the age of the child and their sex. Maternal variables included the age, educational level, depression, and anxiety level of the mothers. Household size, monthly income, and food security status were used as household-level variables. Household food insecurity was assessed using the Household Food Insecurity Access Scale (HFIAS) guideline (version 3) [31]. HFIAS measures the level of food insecurity experienced by households over the past 4 weeks, focusing on their access to food. From the HFIAS, the households were classified as food secure, mild food insecure, moderate food insecure, and severe food insecure. Later, for the bivariate and multivariate analysis, the households were classified into two groups, food secure and food insecure (merging mild, moderate, and severe food insecure).

## Statistical analysis

Child nutrition status, child feeding index, maternal anxiety and depression, and other background characteristics were summarized as frequencies and percentages. All the independent variables were recorded into 2 or 3 groups. Household income was categorized as 'Low' and 'High' based on computation of the $50^{th}$ percentile. A chi-square test was run to find the association between child undernutrition (stunting, underweight, and wasting) and other individual and household-level background characteristics. The association between Maternal mental health (anxiety and depression) and socio-demographic variables was also checked. Multivariate logistic regression models were built to find the relationship of child undernutrition (stunting, underweight, and wasting) with maternal mental health (anxiety and depression), child feeding index, and other risk factors.

## Regression model building

Three separate logistic regression models (taking into account the dependent variables to be (i) stunting, (ii) wasting, and (iii) underweight) have been carried out to estimate the impact of maternal mental health and other risk factors on a child's nutritional status. Variables with significant associations at p-value <0.05 in bivariate analyses were considered for inclusion in the regression models [32]. All the final model choices were calculated using the Hosmer and Lemeshow goodness-of-fit test, and the significance of the parameters was reviewed using the Wald test. The estimates of the

strengths of associations were exhibited by the adjusted odds ratio (AOR) with a 95% confidence interval (CI). A p-value of 0.05 was considered to be statistically significant. The model's multicollinearity was checked using the correlation of the independent variables and also the variance inflation factor (VIF) to remove the bias or confounder. IBM Statistical Packages for Social Science (SPSS), version 21, was used to perform all statistics.

### Ethics statement

The nature and goal of the study were thoroughly explained to the mother of the children before the interview. Both oral and written informed consent were obtained from the mother of each study participant. Ethical approval for the study was obtained from the Institutional Review Board (IRB) of the Faculty of Biological Sciences, University of Dhaka (Ref. No. 238/Biol. Scs).

## Result

### Characteristics of the study subject

Table 1 presents the background characteristics of the study participants. All the children were under two years old, and most mothers were younger than 25 years. A significant proportion of mothers were homemakers, had only primary-level education, and belonged to low-income households. More than 80% of households reported some degree of food insecurity. The prevalence of child undernutrition was notable, with stunting being the most common form. More than half of the children had a high child feeding index score, while maternal underweight was observed in a small proportion. Regarding maternal mental health, nearly half of the mothers experienced some level of depression (54.9%), and anxiety (50.1%) was prevalent among a similar proportion. Further details on these characteristics are provided in Table 1.

### Bivariate associations between sociodemographic characteristics and child nutritional status

Table 2 displays the relationships between child, household, and maternal characteristics with child nutritional status such as height for age Z-score, weight for height Z-score, and weight for age Z-score. Stunting in children was substantially correlated with child sex. Compared to female children (38.7%), male children had a lower prevalence of stunting (25.3%). Maternal age and education level were linked to stunting in children. Stunting in children was more common in mothers who were younger (36.9%) and in mothers with lower levels of education (36.3%). Household occupation and family income were also linked to stunting in children. Maternal nutritional status (BMI) was found to be significant in children underweight. Household food security, child feeding index, maternal depression, and maternal anxiety were significantly associated with child stunting, wasting, and underweight. Child stunting was more than two times higher with a low score on the IYCF (44.7%) as compared to a high IYCF score (19.5%). Child stunting, wasting, and being underweight were about two and three times more common with low scores on the IYCF as compared to high IYCF scores, respectively. The proportion of stunting, wasting, and underweight children of mothers with depression or anxiety was considerably higher than that of children whose mothers did not experience these conditions.

### Bivariate associations between sociodemographic characteristics and maternal mental health

The associations of maternal anxiety and depression with sociodemographic variables are shown in Table 3. Family income, household food security, and child feeding index score were found to be significantly associated with maternal anxiety and depression. Maternal age, education status, occupation, and Body Mass Index were not found any significant link with maternal mental health. Maternal anxiety and depression were more common among low-income households, 59.2%, and 60.7% respectively. In contrast to households experiencing moderate to severe food insecurity, those with food security or mild food insecurity were less likely to have maternal anxiety and depression. Poor feeding practices (low child feeding index score) were more prevalent among mothers with anxiety (56.0%) than those with no anxiety (44.0%).

**Table 1. Sociodemographic, nutritional, and maternal mental health characteristics of the study subject.**

| Sociodemographic characteristics | n (%) | Sociodemographic characteristics | n (%) |
|---|---|---|---|
| **Age of the children (months)** | | **Household food insecurity** | |
| 6-11 | 158 (39.8) | Food secure | 73 (18.4) |
| 12-17 | 118 (29.7) | Mild food insecure | 125 (31.5) |
| 18-23 | 121 (30.5) | Moderate food insecure | 93 (23.4) |
| **Sex of the children** | | Severe food insecure | 106 (26.7) |
| Male | 199 (50.1) | **Maternal nutritional status (BMI)** | |
| Female | 198 (49.9) | Underweight (BMI: < 18.5) | 51 (12.8) |
| **Maternal age (years)** | | Normal/overweight (BMI: ≥ 18.5) | 346 (87.2) |
| 15-24 | 226 (56.9) | **Child Nutritional Status** | |
| 25-41 | 171 (43.1) | Underweight (WAZ: ≤ -2.0 SD) | 95 (24.1) |
| **Maternal occupation** | | Stinting (HAZ: ≤ -2.0 SD) | 125 (31.9) |
| Homemaker | 338 (85.1) | Wasting (BMI Z score: ≤ -2.0 SD) | 55 (14.0) |
| Others (garment workers/day laborers) | 59 (14.9) | **Child Feeding Index** | |
| **Maternal education** | | Low (score 1–4) | 175 (44.1) |
| Primary & below | 229 (57.7) | High (score 5–7) | 222 (55.9) |
| Secondary or higher | 168 (42.3) | **Maternal Depression level** | |
| **Household size** | | No depression (PHQ: score 0–4) | 179 (45.1) |
| ≤4 members | 276 (69.5) | Mild depression (PHQ: score 5–9) | 152 (38.3) |
| ≥5 members | 121 (30.5) | Moderate depression (PHQ: score 10–14) | 53 (13.4) |
| **Father's occupation** | | Severe depression (PHQ: score 15–27) | 13 (3.3) |
| Laborer or rickshaw puller | 220 (55.4) | **Maternal Anxiety level** | |
| Small business or service holder | 177 (44.6) | No anxiety (GAD-7: score 0–4) | 198 (49.9) |
| **Household income (BDT)** | | Mild anxiety (GAD-7: score 5–9) | 133 (33.5) |
| Low (< 13000) | 196 (49.4) | Moderate anxiety (GAD-7: score 10–14) | 58 (14.6) |
| High (≥ 13000) | 201 (50.6) | Severe anxiety (GAD-7: score 15–21) | 8 (2.0) |

Like anxiety, poor feeding practices were also more prevalent among mothers with depression (63.4.0%) than those with no depression (36.6%).

**Multivariate analyses of the relationship between maternal mental health and other risk factors for child stunting, wasting, and underweight**

Table 4 depicts the findings from binary regression models examining the relationship of maternal mental health and other risk factors with child undernutrition. Maternal mental health was assessed by depression and anxiety levels. Mothers with high levels of depression had 1.8 times higher chance of having a stunted child compared to mothers without depression when controlling for child, maternal, and household characteristics (AOR=1.80, 95% CI=1.10–2.94, p<0.05). While there was a significant correlation in bivariate analysis, multivariate regression analysis could not find any significant association between mother anxiety and stunting in children (Table 4). After controlling for the impact of other factors in the pooled data set, mothers with depression had a roughly three-fold increased likelihood of having wasted children compared to mothers without depression (AOR=2.70, 95% CI=1.38–5.28, p<0.05). Mothers who experienced anxiety had a nearly two-fold increased risk of having underweight children compared to mothers without anxiety (AOR=1.77, 95% CI=1.04–3.11, p<0.05). After controlling other sociodemographic variables, there was no longer any discernible link between underweight and maternal depression.

In multivariate regression analysis, the child's gender, the mother's education, the father's employment, and the child feeding index were additional potential predictors of stunting, similar to maternal depression (Table 4). Compared to boys,

**Table 2. Association between background characteristics of the sample and child nutrition.**

| Characteristics | Height-for-age Z-score | | BMI-for-age -Z score | | Weight-for-age Z-score | |
|---|---|---|---|---|---|---|
| | Norma (n = 276) | Stunting (n = 125) | Normal (n = 337) | Wasting (n = 55) | Normal (n = 300) | Underweight (n = 95) |
| | n (%) | n (%) | n (%) | n (%) | n (%) | n (%) |
| **Sex of the children** | | | | | | |
| Male | 148 (74.7) | 50 (25.3) | 167 (84.3) | 31 (15.7) | 158 (79.8) | 40 (20.2) |
| Female | 119 (61.3) | 75 (38.7) | 170 (87.6) | 24 (12.4) | 142 (72.1) | 55 (27.9) |
| *p-value* | <0.01 | | >0.05 | | >0.05 | |
| **Age of the children (months)** | | | | | | |
| 6-11 | 111 (71.2) | 45 (28.8) | 136 (87.2) | 20 (12.8) | 124 (79.0) | 33 (21.0) |
| 12-17 | 80 (68.4) | 37 (31.6) | 101 (86.3) | 16 (13.7) | 88 (75.2) | 29 (24.8) |
| 18-23 | 76 (63.9) | 43 (36.1) | 100 (84.0) | 19 (16.0) | 88 (72.7) | 33 (27.3) |
| *p-value* | >0.05 | | >0.05 | | >0.05 | |
| **Maternal age (years)** | | | | | | |
| 15-24 | 142 (63.1) | 83 (36.9) | 197 (87.6) | 28 (12.4) | 168 (74.3) | 58 (25.7) |
| 25-41 | 125 (74.9) | 42 (25.1) | 140 (83.8) | 27 (16.2) | 132 (78.1) | 37 (21.9) |
| *p-value* | <0.05 | | >0.05 | | >0.05 | |
| **Maternal occupation** | | | | | | |
| Homemaker | 231 (69.0) | 104 (31.0) | 288 (86.0) | 47 (14.0) | 258 (76.6) | 79 (23.4) |
| Others (garment workers/day laborers) | 36 (63.2) | 21 (36.8) | 49 (86.0) | 8 (14.0) | 42 (72.4) | 16 (27.6) |
| *p-value* | >0.05 | | >0.05 | | >0.05 | |
| **Maternal education** | | | | | | |
| Primary & below | 144 (63.7) | 82 (36.3) | 192 (85.0) | 34 (15.0) | 165 (72.7) | 62 (27.3) |
| Secondary or higher | 123 (74.1) | 43 (25.9) | 145 (87.3) | 21 (12.7) | 135 (80.4) | 33 (19.6) |
| *p-value* | <0.05 | | >0.05 | | >0.05 | |
| **Father's occupation** | | | | | | |
| Laborer or rickshaw puller | 135 (61.6) | 84 (38.4) | 189 (86.3) | 30 (13.7) | 167 (75.9) | 53 (24.1) |
| Small business or service holder | 132 (76.3) | 41 (23.7) | 148 (85.5) | 25 (14.5) | 133 (76.0) | 42 (24.0) |
| *p-value* | <0.01 | | >0.05 | | >0.05 | |
| **Household income (BDT)** | | | | | | |
| Low (< 13000) | 118 (60.5) | 77 (39.5) | 164 (84.1) | 31 (15.9) | 145 (74.4) | 50 (25.6) |
| High (≥13000) | 149 (75.6) | 48 (24.4) | 173 (87.8) | 24 (12.2) | 155 (77.5) | 45 (22.5) |
| *p-value* | <0.01 | | >0.05 | | >0.05 | |
| **Maternal nutritional status (BMI)** | | | | | | |
| Underweight (BMI < 18.5) | 34 (68.0) | 16 (32.0) | 42 (84.0) | 8 (16.0) | 32 (64.0) | 18 (36.0) |
| Normal/overweight (BMI ≥ 18.5) | 233 (68.1) | 109 (31.9) | 295 (86.3) | 47 (13.7) | 268 (77.7) | 77 (22.3) |
| *p-value* | >0.05 | | >0.05 | | <0.05 | |
| **Household food insecurity** | | | | | | |
| Food secure/mild insecure | 145 (74.4) | 50 (25.6) | 175 (89.7) | 20 (10.3) | 164 (82.8) | 34 (17.2) |
| Moderate/severe food insecure | 122 (61.9) | 75 (38.1) | 162 (82.2) | 35 (17.8) | 136 (69.0) | 61 (31.0) |
| *p-value* | <0.01 | | <0.05 | | <0.01 | |
| **Child Feeding Index** | | | | | | |
| Low (score 1–4) | 90 (52.3) | 82 (44.7) | 140 (81.4) | 32 (18.6) | 105 (60.7) | 68 (39.3) |
| High (score 5–7) | 177 (80.5) | 43 (19.5) | 197 (89.5) | 23 (10.5) | 195 (87.8) | 27 (12.2) |
| *p-value* | <0.001 | | <0.05 | | <0.001 | |

*(Continued)*

 

**Table 2.** (Continued)

| Characteristics | Height-for-age Z-score | | BMI-for-age -Z score | | Weight-for-age Z-score | |
|---|---|---|---|---|---|---|
| | Norma (n=276) | Stunting (n=125) | Normal (n=337) | Wasting (n=55) | Normal (n=300) | Underweight (n=95) |
| | n (%) | n (%) | n (%) | n (%) | n (%) | n (%) |
| **Maternal Depression level** | | | | | | |
| No depression (PHQ: score 0–4) | 136 (76.4) | 42 (23.6) | 166 (92.7) | 13 (7.3) | 146 (82.0) | 32 (18.0) |
| Depression (PHQ: score 5–27) | 131 (61.2) | 83 (38.8) | 172 (80.4) | 42 (19.6) | 154 (71.0) | 63 (29.0) |
| *p-value* | <0.01 | | <0.001 | | <0.01 | |
| **Maternal Anxiety level** | | | | | | |
| No anxiety (GAD-7: score 0–4) | 145 (74.0) | 51 (26.0) | 176 (89.8) | 20 (10.2) | 164 (82.8) | 34 (17.2) |
| anxiety (GAD-7: score 5–21) | 122 (62.2) | 74 (37.8) | 161 (82.1) | 35 (17.9) | 136 (69.0) | 61 (31/0) |
| *p-value* | <0.05 | | <0.05 | | <0.05 | |

female children had a roughly two-fold higher risk of stunting (AOR = 2.13, CI = 1.32–3.44, p < 0.01). Children of younger mothers had nearly double the risk of stunting (AOR = 1.97, CI = 1.20–3.22, p < 0.01) compared to older mothers. Child-hood stunting was more common in laborer or rickshaw puller household heads families (AOR = 1.84, CI: 1.10–3.08, p < 0.05). A lower child feeding index score was associated with a higher risk of stunting (AOR = 3.21, CI: 1.99–5.16, p < 0.001) and underweight (AOR = 4.20, CI: 2.50–7.07, p < 0.01) in children. Compared to normal/overweight mothers, the children of an underweight mother had almost twice the chance of becoming underweight (AOR = 2.01, CI = 1.01–4.03, p < 0.05).

## Discussion

The present study explores the prevalence of child undernutrition in low-income settings of Dhaka city. It also highlights the maternal mental health and other sociodemographic factors of child undernutrition.

### Child undernutrition and maternal mental health

In the present study, the prevalence of stunting, wasting, and being underweight were 31.9%, 14%, and 24.1%, respectively, in low-income regions of Dhaka. Although Bangladesh has achieved tremendous success in combating child under-nutrition over the past decades, the prevalence of stunting, wasting, and underweight remains at 24%, 11%, and 24%, respectively at the national level [3]. Child nutrition is generally poorer in rural and urban slum areas, with previous studies reporting higher rates of stunting, wasting, and underweight among children in urban slum areas [9,19]. Several underlying factors, including maternal physical and mental health, have been identified as key contributors to this elevated burden of child undernutrition [9,19].

The present study assessed maternal mental health by depression and anxiety levels by PHQ-9 and GAD-7, respectively. The PHQ-9 is a self-administered adaptation of the PRIME-MD diagnostic instrument, specifically designed to assess common mental disorders (CMD) [20]. Beyond its role in screening and diagnosing depressive disorders based on specific criteria, the PHQ-9 also stands out as a reliable and valid tool for assessing the severity of depression. Similarly, previous study provides robust support for the GAD-7 as an efficient and valid self-report measure of anxiety [21]. Further-more, it underscores the measure's applicability in evaluating anxiety symptoms across diverse and heterogeneous sam-ples. According to the PHQ-9, more than half of mothers experienced a level of depression (38.3% with mild depression, 13.4% with moderate depression, and 3.3% with severe depression) in the present study. Similarly, based on the GAD-7 scale, half of the mothers experienced a level of anxiety; 33.3% were suffering from mild anxiety, 14.6% were suffering from moderate anxiety, and 2.0% were suffering from severe anxiety. This result is comparable to previous studies in

**Table 3. Association between background characteristics of the sample and maternal mental health.**

| Characteristics | Anxiety | | | Depression | | |
|---|---|---|---|---|---|---|
| | Absent (n = 198) | Present (n = 199) | *p-value* | Absent (n = 179) | Present (n = 218) | *p-value* |
| | n (%) | n (%) | | n (%) | n (%) | |
| **Maternal age (years)** | | | | | | |
| 15-24 | 116 (51.3) | 110 (48.7) | >0.05 | 102 (45.1) | 124 (54.9) | >0.05 |
| 25-41 | 82 (48.0) | 89 (52.0) | | 77 (45.0) | 94 (55.0) | |
| **Maternal occupation** | | | | | | |
| Homemaker | 168 (49.7) | 170 (50.3) | >0.05 | 153 (45.3) | 185 (54.7) | >0.05 |
| Others (garment workers/day laborers) | 30 (50.8) | 29 (49.2) | | 26 (44.1) | 33 (55.9) | |
| **Maternal education** | | | | | | |
| Primary & below | 105 (45.9) | 124 (54.1) | >0.05 | 98 (42.8) | 131 (57.2) | >0.05 |
| Secondary or higher | 93 (55.4) | 75 (44.6) | | 81 (48.2) | 87 (51.8) | |
| **Father's occupation** | | | | | | |
| Laborer or rickshaw puller | 109 (49.5) | 111 (50.5) | >0.05 | 97 (44.1) | 123 (55.9) | >0.05 |
| Small business or service holder | 89 (50.3) | 88 (49.7) | | 82 (46.3) | 95 (53.7) | |
| **Household income (BDT)** | | | | | | |
| Low (< 13000) | 80 (40.8) | 116 (59.2) | <0.001 | 77 (39.3) | 119 (60.7) | <0.05 |
| High (≥13000) | 118 (58.7) | 83 (41.3) | | 102 (50.7) | 99 (49.3) | |
| **Maternal nutritional status (BMI)** | | | | | | |
| Underweight (BMI: < 18.5) | 26 (51.0) | 25 (49.0) | >0.05 | 20 (39.2) | 31 (60.8) | >0.05 |
| Normal/overweight (BMI: ≥ 18.5) | 172 (49.7) | 174 (50.3) | | 159 (46.0) | 187 (54.0) | |
| **Household food insecurity** | | | | | | |
| Food secure/mild insecure | 129 (65.2) | 69 (34.8) | <0.001 | 102 (51.5) | 96 (48.5) | <0.05 |
| Moderate/severe food insecure | 69 (34.7) | 130 (65.3) | | 77 (38.7) | 122 (61.3) | |
| **Child Feeding Index** | | | | | | |
| Low (score 1–4) | 77 (44.0) | 98 (56.0) | <0.05 | 64 (36.6) | 111 (63.4) | <0.01 |
| High (score 5–7) | 121 (54.5) | 101 (45.5) | | 115 (51.8) | 107 (48.2) | |

Bangladesh [9,13]. They found 46.2–49% of Bangladeshi mothers with CMD. Maternal CMD is more common in under-developed nations: Vietnam (31.2%), Ethiopia (39.5%), Peru (30%), and India (30.0%) [12]. Several potential causes for higher maternal mental health problems, including poorer socioeconomic status, food poverty, age, illiteracy, undernutrition, unsupportive spouses, and physical abuse, were mentioned in previous studies [8,9,13]. Hence, in the present study, most of the mothers were less educated and from food-insecure households, which justified the higher prevalence of their depression and anxiety levels.

### Maternal mental health as a risk factor for child undernutrition

The present study explored the association of maternal depression and anxiety with the undernutrition of children. According to the study findings, mothers with high levels of depression had a significantly greater likelihood of having stunted and wasted children, even after adjusting for child, maternal, and household characteristics. Similarly, maternal anxiety was associated with a higher risk of underweight in children, but its link to stunting was not significant in the multivariate analysis. Previous studies also identified maternal CMD as a risk factor for their child's undernutrition [8,9,12,13,15]. The studies reported that when maternal CMD existed the children were found more likely to be stunted, wasted, and underweight.

**Table 4. Binary regression Model for the determinant of child stunting, underweight, and wasting.**

| | Model I Stunted AORs (95% CI) | Model II Underweight AORs (95% CI) | Model III Wasted AORs (95% CI) |
|---|---|---|---|
| **Sex of children** | | | |
| Male (r) | 1 | | |
| Female | 2.13 (1.32, 3.44) ** | | |
| **Maternal age (years)** | | | |
| 25-41 (r) | 1 | | |
| 15-24 | 1.97 (1.20, 3.22) ** | | |
| **Maternal education** | | | |
| Secondary or higher (r) | 1 | | |
| Primary & below | 1.61 (0.98, 2.66) | | |
| **Maternal BMI** | | | |
| Normal/overweight (BMI: ≥ 18.5) (r) | | 1 | |
| Underweight (BMI: < 18.5) | | 2.01 (1.01, 4.03) * | |
| **Father's occupation** | | | |
| Small business or service holder (r) | 1 | | |
| Laborer or rickshaw puller | 1.84 (1.10, 3.08) * | | |
| **Household income (BDT)** | | | |
| High (≥13000) (r) | 1 | | |
| Low (< 13000) | 1.17 (0.68, 2.01) | | |
| **Household food insecurity** | | | |
| Food secure/mild insecure (r) | 1 | 1 | 1 |
| Moderate/severe food insecure | 1.11 (0.65, 1.90) | 1.51 (0.89, 2.58) | 1.40 (0.75, 2.64) |
| **Child Feeding Index** | | | |
| High (score 5–7) (r) | 1 | 1 | 1 |
| Low (score 1–4) | 3.21 (1.99, 5.16) *** | 4.20 (2.50, 7.07) ** | 1.58 (0.87, 2.88) |
| **Maternal Anxiety level** | | | |
| No anxiety (GAD-7: score 0–4) (r) | 1 | 1 | 1 |
| anxiety (GAD-7: score 5–21) | 1.51 (0.92, 2.48) | 1.77 (1.04, 3.01) * | 1.56 (0.83, 2.92) |
| **Maternal Depression level** | | | |
| No depression (PHQ: score 0–4) (r) | 1 | 1 | 1 |
| Depression (PHQ: score 5–27) | 1.80 (1.10, 2.94) * | 1.40 (0.84, 2.36) | 2.70 (1.38, 5.28) * |

*Indicate the significant at *p-value* <0.05; **Indicate the significant at *p-value* <0.01; ***Indicate the significant at *p-value* <0.001;

Abbreviations: AOR, adjusted odds ratio; CI, confidence interval; HH, household; BMI, body mass index; HFIAS, Household food insecurity access scale

The relationship between maternal mental health and child undernutrition has been explained in different pathways in previous studies. Two significant predictors of optimum growth and development of the children are appropriate feeding practices and proper care [10]. A depressed mother with high CMD might not properly take care of herself and her children [11–13,15]. Thus, due to poor care and hygiene practices, the frequent occurrence of diarrhea and acute respiratory infections (ARI) were found among the children of mothers with depression and anxiety. On the other hand, frequent attacks of diarrhea and ARI hamper the appetite of the children and subsequently lead them to acute undernutrition like wasting and being underweight. A bidirectional association has been reported in a previous study; acute diseases (diarrhea, ARI), less appetite, wasting, etc., might affect maternal mental health. Moreover, the feeding practices of the children are also adversely affected by maternal mental health, resulting in child undernutrition like wasting in the acute phase and stunting in the chronic phase [16].

## Sociodemographic risk factors of child undernutrition

In addition to maternal mental health, the present study also identified sociodemographic determinants of child undernutrition. Along with maternal depression, multivariate regression analysis has revealed that the child's sex, the mother's age, the father's occupation, and the child's feeding index are also potential predictors of stunting. In the current study, female children faced approximately twice the risk of stunting compared to males. Several studies indicated that the prevalence of wasting and underweight was higher among female children compared to males [29,33]. However, it's important to note that while some studies suggest male children are more likely to experience undernutrition [6,34], a study conducted in urban slum areas highlighted that female children in such settings are more vulnerable to undernutrition than males [33]. In some communities, there is a belief that male children are more valued, a perception that is often linked to their socio-economic status [35]. As a result, male children are typically given more nutritious food compared to female children, leading to the assumption that they have better nutritional health. Consequently, there is a need to explore the underlying causes behind this disparity, particularly among female children from urban slums and low-income urban areas in Bangladesh.

Another critical determinant of child undernutrition was the child feeding index: children with lower scores on this index were more likely to be both stunted and underweight. Previous studies also highlighted that the breastfeeding and complementary feeding practices of the children influence the nutritional status of the children [30,36,37]. Lower adherence to appropriate breastfeeding, dietary diversity, meal frequency, and acceptable diet contribute to a higher prevalence of stunting, wasting, and underweight. Moreover, feeding practices to the children are adversely affected by maternal mental health [16]. Our study findings corroborate this relationship as they highlighted the concurrent adverse effect of maternal mental health and lower child feeding index on child nutrition.

Additionally, according to the findings of the present study, children of younger mothers were more likely to experience stunting compared to those born to older mothers. Previous studies [38,39] also supported this finding; children born to younger mothers (typically defined as adolescents or women in their early twenties) face a higher risk of stunting. Specifically, those born to mothers aged 19 years or younger exhibit increased vulnerability to low birth weight, preterm birth, and stunting at 2 years of age. The explanation for this correlation can be found in the fact that younger mothers often experience challenges related to their health, education, and socioeconomic status, which can indirectly impact their ability to provide optimal care for their children. Similarly, in our results, children of underweight mothers had nearly double the risk of becoming underweight themselves when compared to babies born to normal or overweight mothers. Previous studies also highlighted that the children of underweight mothers experienced a higher risk of undernutrition compared to the children of normal/overweight mothers [27,29].

Furthermore, households led by laborers or rickshaw pullers showed a higher prevalence of childhood stunting. Evidence suggests that children of poor households or lower-income quintiles are at risk of experiencing undernutrition [26,28,29]. The households led by day laborers or rickshaw pullers are usually of lower-income communities compared to those led by businessmen or service holders. Moreover, these communities have less education and socioeconomic challenges which hinder their ability to feed the family adequately and take care of health and hygiene issues.

## Limitations of the study

The study has some limitations. Firstly, the participants were selected from two regions (police stations) of south and north city corporations, in Dhaka. The participants from these areas might not represent the lower-income regions of Dhaka city. This might restrict us from generalizing the findings for the lower-income population of Dhaka city. A robust study representing the lower income setting of the city could be conducted to point out more generalizable insights on maternal mental health and child undernutrition. Secondly, depression and anxiety were diagnosed by PHQ-9 and GAD-7 scales respectively. There might be recall bias as these scales are based on the self-reported experience two weeks before the data collection.

### Future research directions and implications

This study highlights the association between maternal mental health and child undernutrition, emphasizing the need for further research to explore causal relationships. Future studies should adopt longitudinal designs to determine how maternal depression and anxiety influence child growth over time. Additionally, qualitative research could provide insights into how maternal mental health affects child-feeding practices and nutritional outcomes.

From a policy perspective, integrating maternal mental health support into existing maternal and child health programs is essential. Routine screening for depression and anxiety during antenatal and postnatal care could help identify at-risk mothers early. Community-based interventions promoting mental health awareness, social support, and nutritional education should be prioritized to improve both maternal well-being and child nutrition. Targeted nutrition programs are also needed to address the high prevalence of child undernutrition. Special attention should be given to vulnerable groups, such as young mothers and female children, who were identified as being at higher risk. A multi-sectoral approach, combining mental health support, improved feeding practices, and social protection measures, could help reduce the burden of undernutrition in low-income urban settings.

## Conclusion

The study findings highlighted that around half of the mothers experienced a level of depression and anxiety. A significant portion of the children were found to be stunted, wasted, and underweight. Maternal depression and anxiety adversely affect the nutrition of their children. The children of the mothers with high depression and anxiety were at higher risk of stunting, wasting, and underweight. Besides, different sociodemographic factors, including child's sex, maternal age, maternal health status, and child feeding index, were significant factors of child undernutrition in lower-income regions of Dhaka city. Given this context, policy design should prioritize maternal mental health, child feeding practices, and addressing child undernutrition among mothers in this setting.

## Supporting information

**S1 Table. Supporting data for the results presented in the study.**
(DOCX)

**S1 Appendix. Supporting data for the results presented in the study.**
(XLSX)

## Author contributions

**Conceptualization:** Kazi Muhammad Rezaul karim, Tasmia Tasnim, Sumaiya Akter.

**Data curation:** Md Hafizul Islam, Tasmia Tasnim, Sumaiya Akter.

**Formal analysis:** Kazi Muhammad Rezaul karim, Md Hafizul Islam.

**Funding acquisition:** Kazi Muhammad Rezaul karim.

**Investigation:** Sumaiya Akter.

**Methodology:** Kazi Muhammad Rezaul karim, Tasmia Tasnim.

**Project administration:** Kazi Muhammad Rezaul karim.

**Resources:** Md Hafizul Islam.

**Supervision:** Kazi Muhammad Rezaul karim, Tasmia Tasnim, Sumaiya Akter.

**Visualization:** Kazi Muhammad Rezaul karim.

**Writing – original draft:** Md Hafizul Islam.

**Writing – review & editing:** Kazi Muhammad Rezaul karim, Tasmia Tasnim, Sumaiya Akter.

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
