## [Decision Letter · Decision Letter 0]

3 Mar 2025

Dear Dr. karim,

Thank you for submitting your manuscript to PLOS ONE. After careful consideration, we feel that it has merit but does not fully meet PLOS ONE’s publication criteria as it currently stands. Therefore, we invite you to submit a revised version of the manuscript that addresses the points raised during the review process.

We look forward to receiving your revised manuscript.

Kind regards,

Dinaol Abdissa Fufa, Mph

Academic Editor

PLOS ONE

“University Grant Commission (UGC), Bangladesh support the research grant (2022-23) to public university teacher.  ”

4. We note that your Data Availability Statement is currently as follows: “All relevant data are within the manuscript and in Supporting Information files.”

Reviewers' comments:

Reviewer's Responses to Questions

**Comments to the Author**

1. Is the manuscript technically sound, and do the data support the conclusions?

Reviewer #1: Yes

Reviewer #2: Yes

Reviewer #3: Yes

2. Has the statistical analysis been performed appropriately and rigorously?

Reviewer #1: Yes

Reviewer #2: Yes

Reviewer #3: Yes

3. Have the authors made all data underlying the findings in their manuscript fully available?

Reviewer #1: Yes

Reviewer #2: Yes

Reviewer #3: Yes

4. Is the manuscript presented in an intelligible fashion and written in standard English?

Reviewer #1: Yes

Reviewer #2: No

Reviewer #3: Yes

Reviewer #1: Interesting idea of this study, my recommendations are the following:

Lines 22-25 I recommend to write as a single sentence, for example - assessment of malnutrition status and associated characteristics in children.

I recommend expanding the Introduction section by mentioning the main aspects regarding the mental health of mothers and the risk of associated conditions. Description based on specialized literature of the main mental conditions in breastfeeding mothers.

Line 109 I recommend mentioning the bibliographic index as a number from the bibliographic references, according to the editing rules.

Assessment of Maternal Mental Health – I recommend mentioning the reliability of the questionnaires by calculating Cronbach's alpha regarding the results of this study.

Characteristics of the study subject – I recommend that when interpreting the results, the information should not be duplicated with those mentioned in table 1, I recommend clarifications.

I recommend that in tables regarding the p-value index, the standard symbol should be mentioned: <, >.

I recommend that future research directions and practical implications be mentioned at the end of the Discussions section.

Reviewer #2: The manuscript presents a well-executed study on an important public health issue, exploring the relationship between maternal mental health and child undernutrition in low-income settings. The use of validated mental health assessment tools (PHQ-9, GAD-7) and standardized anthropometric measurements strengthens the reliability of the findings. Additionally, the application of multivariate logistic regression and appropriate model validation enhances the robustness of the statistical analysis. The study’s conclusions are well-supported by the data, and the policy implications are clear, emphasizing the need for maternal mental health interventions in child nutrition programs.

However, the manuscript has several areas that require improvement. The writing quality needs refinement, as certain sentences are overly complex, making it difficult to follow key findings. For instance, in the Discussion section, some interpretations of the regression results could be rewritten more concisely to improve clarity. Additionally, there is some repetition in how maternal depression and anxiety are discussed across different sections, particularly in the Results and Discussion, which could be streamlined to avoid redundancy. Statistical interpretations should also be made more precise—for example, rather than stating that maternal depression “considerably increases” the odds of stunting, specifying the exact percentage increase based on odds ratios would make the findings more accessible to readers.

Furthermore, the manuscript would benefit from clearer terminology. The terms “malnutrition” and “undernutrition” appear to be used interchangeably, though they have different meanings—malnutrition includes both under- and overnutrition, while this study focuses specifically on undernutrition. Consistently using the correct term would enhance clarity. Similarly, the description of the child feeding index could be expanded slightly to help readers unfamiliar with the methodology understand how scores were calculated.

Overall, this is a valuable study with important findings, but revisions to improve clarity, reduce redundancy, and refine statistical explanations would strengthen the manuscript further.

Reviewer #3: The study titled “Child Undernutrition is Associated with Maternal Mental Health and Other Sociodemographic Factors in Low-Income Settings in Dhaka, Bangladesh” by Karim et al. discusses an important topic. However, to further improve their study, I recommend the following edits:

• Pages 11-13, Introduction Section: Given that this study focuses on maternal health, it is important for the authors to elaborate further on maternal mental health and its impact on child nutrition. While the introduction mentions maternal mental health as a factor, it does not sufficiently explain the mechanisms through which maternal depression or anxiety may lead to child undernutrition. Adding a few lines on why maternal mental health is crucial—such as its influence on feeding practices, healthcare-seeking behavior, and overall child well-being—would strengthen the argument and provide a clearer rationale for its inclusion in the study, especially in the context of Bangladesh.

• Page 12, Lines 78-85: As the authors mention that non-slum low-income areas remain unexplored, they should expand on this point and highlight it more, as it represents the novel aspect of their study. Many studies have already been conducted on maternal health in low- and middle-income countries like Bangladesh, so it is important to explain why non-slum low-income areas need separate attention. Clarifying how this population differs from those previously studied would help strengthen the focus and contribution of this research.

• Page 13, Lines 93-99: The study setting and population are well described; however, the inclusion and exclusion criteria for participants are not mentioned. Clarifying these criteria would enhance the transparency and reproducibility of the study. Were there any specific factors that determined participant selection or exclusion?

• Page 14, Lines 119-125: The methodology for anthropometric measurements is unclear. Were these measurements taken by trained professionals, or were they self-reported? Clarifying this would aid in evaluating the accuracy and reliability of the data.

• Page 15, Lines 127-138: The study provides a clear description of the PHQ-9 and GAD-7 scales, but it is unclear whether they have been validated for use in the Bangladeshi population. Given that cultural differences can influence responses to mental health questionnaires, providing this information would strengthen the study’s validity.

• Page 31, Lines 365-378: The authors imply that socio-cultural preferences for male children lead to better nutrition but do not provide direct evidence. Citing studies on gender-based food distribution would strengthen this claim. Adding references would enhance clarity and support the argument.

**Do you want your identity to be public for this peer review?** For information about this choice, including consent withdrawal, please see our Privacy Policy

Reviewer #1: **Yes: ** Badau Adela

Reviewer #2: No

Reviewer #3: No

---

## [Author Response · Author response to Decision Letter 1]

20 Mar 2025

Author’s response to the review comments

Reviewer #1: Interesting idea of this study, my recommendations are the following:

Comment: Lines 22-25 I recommend to write as a single sentence, for example - assessment of malnutrition status and associated characteristics in children.

Author’s response: Thank you for your suggestion. We have revised the sentences in Lines 22-25 to present the information concisely in a single sentence, as recommended. The revised sentence now reads: "This study assesses child undernutrition and its associated characteristics, including maternal mental health, in low-income settings in Dhaka, Bangladesh." See the lines 21-22 in the revised manuscript.

Comment: I recommend expanding the Introduction section by mentioning the main aspects regarding the mental health of mothers and the risk of associated conditions. Description based on specialized literature of the main mental conditions in breastfeeding mothers.

Author’s response: Thank you for your thoughtful comment. We recognize the importance of further elaborating on the mechanisms through which maternal mental health influences child nutrition to strengthen the rationale for our study.

To address this, we have expanded the Introduction section to highlight how maternal depression and anxiety can negatively impact feeding practices, healthcare-seeking behavior, and overall child well-being. Maternal mental health issues can lead to reduced responsiveness in caregiving, lower breastfeeding rates, and poor complementary feeding practices, all of which are critical for a child's growth and development. Additionally, mothers experiencing depression or anxiety may have limited energy, motivation, or confidence to ensure proper hygiene, meal preparation, or timely healthcare visits, increasing the risk of child malnutrition and illness. Please see the lines 65-84 in the revised manuscript.

Comment: Line 109 I recommend mentioning the bibliographic index as a number from the bibliographic references, according to the editing rules.

Author’s response: Thank you for your suggestion. We have revised Line 109 to include the bibliographic reference number in accordance with the journal's editing rules. See the revision “A prior study conducted in an urban slum region in Bangladesh [9], where the prevalence of common mental disorders was 46.2% and stunting was 44.3%, we assumed a sample power of 50%” in the revised manuscript. See the lines 120-121 in the revised manuscript.

Comment: Assessment of Maternal Mental Health – I recommend mentioning the reliability of the questionnaires by calculating Cronbach's alpha regarding the results of this study.

Author’s response: Thank you for your valuable suggestion. We have now assessed the reliability of the questionnaires used to measure maternal mental health in our study. The Cronbach’s alpha values were 0.854 for the Generalized Anxiety Disorder 7-Item Scale (GAD-7) and 0.807 for the Patient Health Questionnaire-9 (PHQ-9), indicating good internal consistency. We have incorporated this information in the manuscript accordingly. See the lines 164-165 and 171-172 in the revised manuscript.

Comment: Characteristics of the study subject – I recommend that when interpreting the results, the information should not be duplicated with those mentioned in table 1, I recommend clarifications.

Author’s response: Thank you for your suggestion. We have revised the interpretation of the study subject characteristics to avoid unnecessary duplication of information already presented in Table 1. Instead of repeating detailed numerical data, we now provide a summarized interpretation while referring readers to Table 1 for specifics. See the lines 239-247 in the revised manuscript.

“Table 1 presents the background characteristics of the study participants. The majority of children were under two years old, and most mothers were younger than 25 years. A significant proportion of mothers were homemakers, had only primary-level education, and belonged to low-income households. More than 80% of households reported some degree of food insecurity. The prevalence of child undernutrition was notable, with stunting being the most common form. More than half of the children had a high child feeding index score, while maternal underweight was observed in a small proportion. Regarding maternal mental health, nearly half of the mothers experienced some level of depression, and anxiety was prevalent among a similar proportion. Further details on these characteristics are provided in Table 1.”

Comment: I recommend that in tables regarding the p-value index, the standard symbol should be mentioned: <, >.

Author’s response: Thank you for your suggestion. The p-value index has been corrected according to your suggestion in the tables.

Comment: I recommend that future research directions and practical implications be mentioned at the end of the Discussions section.

Author’s response: Thank you for your insightful suggestion. We have now included a section at the end of the Discussion to highlight future research directions and implications. See the following section in the revised manuscript (Lines 426-441).

Future research directions and implications

This study highlights the association between maternal mental health and child undernutrition, emphasizing the need for further research to explore causal relationships. Future studies should adopt longitudinal designs to determine how maternal depression and anxiety influence child growth over time. Additionally, qualitative research could provide insights into how maternal mental health affects child-feeding practices and nutritional outcomes.

From a policy perspective, integrating maternal mental health support into existing maternal and child health programs is essential. Routine screening for depression and anxiety during antenatal and postnatal care could help identify at-risk mothers early. Community-based interventions promoting mental health awareness, social support, and nutritional education should be prioritized to improve both maternal well-being and child nutrition. Targeted nutrition programs are also needed to address the high prevalence of child undernutrition. Special attention should be given to vulnerable groups, such as young mothers and female children, who were identified as being at higher risk. A multi-sectoral approach, combining mental health support, improved feeding practices, and social protection measures, could help reduce the burden of undernutrition in low-income urban settings.

Reviewer #2: The manuscript presents a well-executed study on an important public health issue, exploring the relationship between maternal mental health and child undernutrition in low-income settings. The use of validated mental health assessment tools (PHQ-9, GAD-7) and standardized anthropometric measurements strengthens the reliability of the findings. Additionally, the application of multivariate logistic regression and appropriate model validation enhances the robustness of the statistical analysis. The study’s conclusions are well-supported by the data, and the policy implications are clear, emphasizing the need for maternal mental health interventions in child nutrition programs.

Comment: However, the manuscript has several areas that require improvement. The writing quality needs refinement, as certain sentences are overly complex, making it difficult to follow key findings. For instance, in the Discussion section, some interpretations of the regression results could be rewritten more concisely to improve clarity. Additionally, there is some repetition in how maternal depression and anxiety are discussed across different sections, particularly in the Results and Discussion, which could be streamlined to avoid redundancy. Statistical interpretations should also be made more precise—for example, rather than stating that maternal depression “considerably increases” the odds of stunting, specifying the exact percentage increase based on odds ratios would make the findings more accessible to readers.

Author’s response: Thank you for your detailed feedback. We acknowledge the need for improving writing clarity, reducing redundancy, and refining statistical interpretations to enhance readability and precision.

To address these concerns, we have made the following revisions:

• Simplified complex sentences in the Discussion section, ensuring that key findings and interpretations of regression results are presented more concisely.

• Reduced repetition in the Results and Discussion sections, particularly regarding maternal depression and anxiety, by consolidating overlapping points to avoid redundancy.

• Refined statistical interpretations by specifying exact percentage increases in odds based on odds ratios rather than using broad terms like "considerably increases." This enhances precision and makes findings more transparent for readers.

See the lines 291-319, 329-332, 355-360 in the revised manuscript.

Comment: Furthermore, the manuscript would benefit from clearer terminology. The terms “malnutrition” and “undernutrition” appear to be used interchangeably, though they have different meanings—malnutrition includes both under- and overnutrition, while this study focuses specifically on undernutrition. Consistently using the correct term would enhance clarity. Similarly, the description of the child feeding index could be expanded slightly to help readers unfamiliar with the methodology understand how scores were calculated.

Author’s response: Thank you for your insightful comment. We acknowledge the distinction between "malnutrition" and "undernutrition" and agree that consistent terminology will enhance clarity. Our study specifically focuses on assessing undernutrition in children, including stunting, wasting, and underweight. To accurately reflect this focus, we have carefully revised the manuscript to ensure that "undernutrition" is used appropriately throughout, replacing "malnutrition" where necessary. We appreciate your suggestion, which has helped improve the precision and consistency of our terminology.

We have expanded the description of the child feeding index in the Methods section to provide greater clarity for readers unfamiliar with the methodology. Specifically, we have detailed how the index was constructed, including the key components assessed, the scoring criteria, and the categorization of feeding practices. We have provided this information as a supplementary table as well. This addition enhances the transparency of our methodology and ensures a better understanding of how the scores were derived.

Variables Children 6-8 months Children 9-23 months

Category Score Category Score

Breastfeeding Yes 2 Yes 2

No 0 No 0

Bottle feeding Yes 0 Yes 0

No 1 No 1

Dietary diversity score during 24-h 0 food groups 0 0 food groups 0

1-3 food groups 1 1-3 food groups 1

≥ 4 food groups 2 ≥ 4 food groups 2

Meal frequency during 24-h 0 meal/day 0 0 meal/day 0

1 meal/day 1 1-2 meal/day 1

≥ 2 meals/day 2 ≥ 3 meals/day 2

Comment: Overall, this is a valuable study with important findings, but revisions to improve clarity, reduce redundancy, and refine statistical explanations would strengthen the manuscript further.

Author’s response: Thank you for your valuable feedback and for recognizing the importance of our study. We appreciate your suggestions to improve clarity, reduce redundancy, and refine statistical explanations to enhance the manuscript.

To address these concerns, we have:

• Refined the writing to improve clarity and ensure a more concise presentation of findings, particularly in the Discussion section.

• Reduced redundancy by streamlining repetitive content, especially in the Results and Discussion sections, to maintain a focused narrative.

• Improved statistical explanations by providing precise interpretations of odds ratios, ensuring that findings are conveyed with clarity and accuracy.

• Clarified the calculation of the child feeding index to help readers unfamiliar with the methodology better understand how the scores were determined.

See the lines 291-319, 329-332, 355-360 in the revised manuscript. Also see the supplementary table 1.

Reviewer #3: The study titled “Child Undernutrition is Associated with Maternal Mental Health and Other Sociodemographic Factors in Low-Income Settings in Dhaka, Bangladesh” by Karim et al. discusses an important topic. However, to further improve their study, I recommend the following edits:

Comment: Pages 11-13, Introduction Section: Given that this study focuses on maternal health, it is important for the authors to elaborate further on maternal mental health and its impact on child nutrition. While the introduction mentions maternal mental health as a factor, it does not sufficiently explain the mechanisms through which maternal depression or anxiety may lead to child undernutrition. Adding a few lines on why maternal mental health is crucial—such as its influence on feeding practices, healthcare-seeking behavior, and overall child well-being—would strengthen the argument and provide a clearer rationale for its inclusion in the study, especially in the context of Bangladesh.

Author’s response: Thank you for your thoughtful comment. We recognize the importance of further elaborating on the mechanisms through which maternal mental health influences child nutrition to strengthen the rationale for our study.

To address this, we have expanded the Introduction section to highlight how maternal depression and anxiety can negatively impact feeding practices, healthcare-seeking behavior, and overall child well-being. Maternal mental health issues can lead to reduced responsiveness in caregiving, lower breastfeeding rates, and poor complementary feeding practices, all of which are critical for a child's growth and development. Additionally, mothers experiencing depression or anxiety may have limited energy, motivation, or confidence to ensure proper hygiene, meal preparation, or timely healthcare visits, increasing the risk of child malnutrition and illness. Please see the lines 65-84 in the revised manuscript.

Comment: Page 12, Lines 78-85: As the authors mention that non-slum low-income areas remain unexplored, they should expand on this point and highlight it more, as it represents the novel aspect of their study. Many studies have already been conducted on maternal health in low- and middle-income countries like Bangladesh, so it is important to explain why non-slum low-income areas need separate attention. Clarifying how this population differs from those previously studied would help strengthen the focus and contribution of this research.

Author’s response: Thank you for your insightful comment. We appreciate the opportunity to elaborate on the significance of studying non-slum low-income areas and how this population differs from those previously studied.

To address this, we have expanded the discussion to highlight that while urban slum populations in Bangladesh have been widely studied regarding child undernutrition, non-slum low-income areas remain underexplored despite facing similar economic hardships. These areas house marginalized populations who may not be as visible in policy discussions, as they are not classified as slum dwellers but still experience limited access to healthcare, social safety nets, and nutritional resources. Their living conditions, employment patterns, and access to services differ from both slum residents and middle-income urban populations, necessitating a focused investigation.

Furthermore, our study provides novel insights by examining the association between maternal mental health and child undernutrition in this specific non-slum low-income context, an area that has received little research attention. We have revised the manuscript to emphasize these points more clearly and strengthen the study's contribution.

“Several studies identified the socio-economic determinants of child undernutrition in Bangladesh at national and subnational levels [5–9, 16]. Urban areas in Bangladesh encompass a diverse population in terms of distinct economic disparity and lifestyle, which might affect the nutrition of children [17]. A few studies were also conducted in urban slum areas

---

## [Decision Letter · Decision Letter 1]

23 Mar 2025

Child undernutrition is associated with maternal mental health and other sociodemographic factors in low-income settings in Dhaka, Bangladesh

PONE-D-24-58026R1

Dear Dr. karim,

We’re pleased to inform you that your manuscript has been judged scientifically suitable for publication and will be formally accepted for publication once it meets all outstanding technical requirements.

Kind regards,

Dinaol Abdissa Fufa, Mph

Academic Editor

PLOS ONE

Reviewers' comments:

Reviewer's Responses to Questions

**Comments to the Author**

Reviewer #1: All comments have been addressed

Reviewer #2: All comments have been addressed

2. Is the manuscript technically sound, and do the data support the conclusions?

Reviewer #1: Yes

Reviewer #2: Yes

3. Has the statistical analysis been performed appropriately and rigorously?

Reviewer #1: Yes

Reviewer #2: Yes

4. Have the authors made all data underlying the findings in their manuscript fully available?

Reviewer #1: Yes

Reviewer #2: Yes

5. Is the manuscript presented in an intelligible fashion and written in standard English?

Reviewer #1: Yes

Reviewer #2: Yes

Reviewer #1: The article entitled Child undernutrition is associated with maternal mental health and other sociodemographic factors in low-income settings in Dhaka, Bangladesh, was revised in accordance with all suggested recommendations.

The introduction has been carefully revised and supplemented with relevant aspects from the specialized literature, in line with the recommendations.

The Methods section has been revised, the calculation of the reliability index is good, which means that this instrument can be applied to the sample of subjects.

Reviewer #2: The authors have made significant improvements to the manuscript, addressing all the reviewers' concerns thoroughly. Key aspects of the study, including the impact of maternal mental health on child nutrition, have been clarified and expanded. Statistical analysis has been appropriately revised to enhance clarity and precision, with odds ratios now expressed in a more interpretable manner. Additionally, terminology has been standardized, and redundant sections have been streamlined, ensuring better readability.

Specific improvements include:

1.Expansion of the introduction to detail the mechanisms by which maternal mental health influences child nutrition.

2.Validation of the PHQ-9 and GAD-7 scales for the Bangladeshi population, along with Cronbach’s alpha values indicating reliability.

3.Inclusion of inclusion/exclusion criteria and clarification of anthropometric measurement procedures.

4.Addressing all technical issues with data presentation, including p-value formatting and avoiding duplication of information in the results and tables.

The manuscript is now clear, concise, and impactful, making it suitable for publication in PLOS ONE. No further revisions are recommended.

**Do you want your identity to be public for this peer review?** For information about this choice, including consent withdrawal, please see our Privacy Policy

Reviewer #1: **Yes: ** Badau Adela

Reviewer #2: No

---

## [Editor Report · Acceptance letter]

PONE-D-24-58026R1

PLOS ONE

Dear Dr. karim,

I'm pleased to inform you that your manuscript has been deemed suitable for publication in PLOS ONE. Congratulations! Your manuscript is now being handed over to our production team.

Kind regards,

on behalf of

Dr. PLOS Manuscript Reassignment

Staff Editor

PLOS ONE